# Crash landing of *Vibrio cholerae* by MSHA pili-assisted braking and anchoring in a viscoelastic environment

**Wenchao Zhang[1†], Mei Luo[2†], Chunying Feng[1], Huaqing Liu[1], Hong Zhang[1], Rachel R Bennett[3*], Andrew S Utada[4,5*], Zhi Liu[2*], Kun Zhao[1*]**

[1]Frontier Science Center for Synthetic Biology and Key Laboratory of Systems Bioengineering (Ministry of Education), School of Chemical Engineering and Technology, Tianjin University, Tianjin, China; [2]Department of Biotechnology, College of Life Science and Technology, Huazhong University of Science and Technology, Wuhan, China; [3]School of Mathematics, University of Bristol, Bristol, United Kingdom; [4]Faculty of Life and Environmental Sciences, University of Tsukuba, Ibaraki, Japan; [5]The Microbiology Research Center for Sustainability, University of Tsukuba, Ibaraki, Japan

**Abstract** Mannose-sensitive hemagglutinin (MSHA) pili and flagellum are critical for the surface attachment of *Vibrio cholerae*, the first step of *V. cholerae* colonization on host surfaces. However, the cell landing mechanism remains largely unknown, particularly in viscoelastic environments such as the mucus layers of intestines. Here, combining the cysteine-substitution-based labeling method with single-cell tracking techniques, we quantitatively characterized the landing of *V. cholerae* by directly observing both pili and flagellum of cells in a viscoelastic non-Newtonian solution consisting of 2% Luria-Bertani and 1% methylcellulose (LB+MC). The results show that MSHA pili are evenly distributed along the cell length and can stick to surfaces at any point along the filament. With such properties, MSHA pili are observed to act as a brake and anchor during cell landing which includes three phases: running, lingering, and attaching. Importantly, loss of MSHA pili results in a more dramatic increase in mean path length in LB+MC than in 2% LB only or in 20% Ficoll solutions, indicating that the role of MSHA pili during cell landing is more apparent in viscoelastic non-Newtonian fluids than viscous Newtonian ones. Our work provides a detailed picture of the landing dynamics of *V. cholerae* under viscoelastic conditions, which can provide insights into ways to better control *V. cholerae* infections in a real mucus-like environment.

**\*For correspondence:**
rachel.bennett@bristol.ac.uk (RRB);
utada.andrew.gm@u.tsukuba.ac.jp (ASU);
zhiliu@hust.edu.cn (ZL);
kunzhao@tju.edu.cn (KZ)

[†]These authors contributed equally to this work

**Competing interests:** The authors declare that no competing interests exist.

## Introduction

*Vibrio cholerae*, a human pathogen that causes the debilitating disease cholera, is a natural inhabitant of aquatic ecosystems (*Almagro-Moreno et al., 2015*; *Kaper et al., 1995*). They can form biofilms on both biotic and abiotic surfaces, which increase their infectivity and environmental survival (*Donlan and Costerton, 2002*; *Silva and Benitez, 2016*; *Teschler et al., 2015*; *Yildiz and Visick, 2009*).

Bacterial appendages have been shown to play important roles in regulating bacterial activities, especially biofilm formation during microbe-host interactions. The flagellum is required for biofilm formation in a variety of bacteria species, such as *Escherichia coli* (*Pratt and Kolter, 1998*), *Pseudomonas aeruginosa* (*O'Toole and Kolter, 1998*), and *V. cholerae* (*Guttenplan and Kearns, 2013*; *Watnick and Kolter, 1999*). Mutants lacking flagella in both *E. coli* and *Vibrio vulnificus* have been observed to be defective for attachment (*Friedlander et al., 2013*; *Lee et al., 2004*). Type IV pili (TFP) are another type of filamentous appendages commonly found on many bacteria and

archaea, which have diverse functions such as cellular twitching motility, biofilm formation, horizontal gene transfer, and host colonization (*Piepenbrink and Sundberg, 2016*). *P. aeruginosa* displays two types of TFP-driven twitching motility (*Gibiansky et al., 2010*). *Neisseria gonorrhoeae* has shown a TFP-dependent attachment, leading to the formation of microcolonies on host cell surfaces (*Higashi et al., 2007*). In contrast, although *V. cholerae* biosynthesizes three types of TFP that are expressed under different scenarios, they have not been observed to twitch on surfaces. These three pili are chitin-regulated competence pili (ChiRP; formerly termed PilA), toxin co-regulated pili (TCP), and mannose-sensitive hemagglutinin (MSHA) TFP (*Meibom et al., 2004*; *Reguera and Kolter, 2005*; *Yildiz and Visick, 2009*). ChiRP were observed to be able to grasp extracellular DNA and transport it back to the cell surface via pili retraction (*Ellison et al., 2018*). TCP are important for host colonization and pathogenesis (*Kirn et al., 2000*; *Thelin and Taylor, 1996*). In contrast to these two types, MSHA pili are known to be important for surface attachment of *V. cholerae* (*Utada et al., 2014*; *Watnick and Kolter, 1999*).

Motility has been shown to be a crucial element for *V. cholerae* colonization of the epithelium, leading to successful infection of the human host (*Krukonis and DiRita, 2003*; *Tsou et al., 2008*). Two types of near-surface motility, roaming and orbiting, were observed in *V. cholera* and were further suggested that *V. cholerae* synergistically employs the use of their flagella and MSHA pili to enable a hybrid surface motility that facilitates surface selection and attachment in aqueous environments (*Utada et al., 2014*). However, there is a lack of direct observational evidence of the appendages in question. Moreover, in addition to the aqueous environments that *V. cholerae* typically inhabits, it also encounters viscoelastic environments in the intestinal mucus of hosts (*Almagro-Moreno et al., 2015*). The mucus layer of animal intestines is estimated to have a wide range of viscosities, varying anywhere from the viscosity of water (~1 cP) to 1000-fold higher (1000 cP) (*Lai et al., 2009*). How cells land on surfaces in such viscoelastic environments is still not clear. To answer these questions, direct live-cell visualization of the pili and flagellum in real-time in viscoelastic conditions is needed.

Recently, there have been significant advances in techniques for directly observing cell appendages (*Blair et al., 2008*; *Ellison et al., 2019*; *Ellison et al., 2018*; *Ellison et al., 2017*; *Nakane and Nishizaka, 2017*; *Renault et al., 2017*; *Skerker and Berg, 2001*; *Talà et al., 2019*). Among them, the cysteine substitution-based labeling method is specific and has been successfully applied to visualize tight adherence (TAD) pili of *Caulobacter crescentus* and TFP of *V. cholera* (*Ellison et al., 2019*).

In this paper, by combining a cysteine substitution-based labeling method with single-cell tracking, we directly observed individual pili and the flagellum of landing cells in viscoelastic media and revealed the dynamic landing sequence of *V. cholerae* as it makes initial surface attachment. The role of MSHA pili during cell landing in a viscoelastic environment is also demonstrated. Our work provides a detailed picture of the landing dynamics of *V. cholerae* under viscoelastic conditions, during which, the synergistic functions of MSHA pili and flagellum are elucidated.

## Results

### MSHA pili are evenly distributed along cell length with a constant length density

To visualize the MSHA pili, we constructed a mutant (MshAT70C) by cysteine substitution, which can subsequently be labeled with highly specific maleimide dyes (*Figure 1a* and *Figure 1—figure supplement 1*), following the protocol in *Ellison et al., 2019*; *Ellison et al., 2017*. To observe the distribution of MSHA pili on the cell surface, we simultaneously stained the plasma membrane with FM4-64 in *Figure 1a*.

We visualized the positions of the different pili as the cell body rotates by recording high-speed videos during surface landing (see one example of a three-dimensional model of a single cell reconstructed from the videos in *Figure 1—figure supplement 2*). The results show evenly distributed MSHA pili along the cell length, indicating the absence of preferred pili localization on the cell body. Quantitatively, we find that the majority of cells have approximately 3–7 MSHA pili, with 4 MSHA pili per cell being observed most frequently, as shown in *Figure 1b*. These results are in agreement with

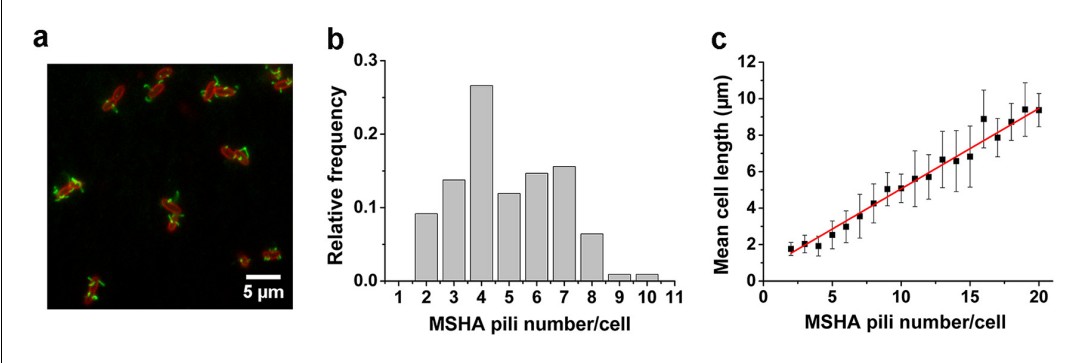

**Figure 1.** MSHA pili are evenly distributed along cell length with a constant length density. (a) Examples of labeled MSHA pili observed on cell bodies. Green fluorescence showing the AF488-mal labeled MSHA pili, red fluorescence showing the FM4-64 labeled plasma membrane. (b) Distribution of pili number per cell cultivated in LB medium. $N_{cell}$=110. (c) The MSHA pili number per cell is linearly correlated with the cell length. Cells with longer length were obtained by 30–50 min treatment using 10 μg/mL cephalexin. $N_{cell}$=368. LB, Luria-Bertani; MSHA, mannose-sensitive hemagglutinin.

The online version of this article includes the following source data and figure supplement(s) for figure 1:

**Source data 1.** Source data for *Figure 1b*.

**Source data 2.** Source data for *Figure 1c*.

**Figure supplement 1.** Labeling of *Vibrio cholerae* MSHA pilus protein MshA with AF488-mal.

**Figure supplement 2.** A 3D view of a typical *Vibrio cholerae* cell showing the whole-body distribution of MSHA pili; this cell has six pili. MSHA, mannose-sensitive hemagglutinin.

**Figure supplement 3.** MSHA pili labeling of the prepared samples.

**Figure supplement 4.** Hemagglutination assays.

recent reports (*Floyd et al., 2020*). Under our conditions, we observed MSHA pili growth (*Figure 1—figure supplement 3a and b*) but no retraction.

The number of MSHA pili appears to be positively correlated with cell length since it increases as the cell grows (*Figure 1—figure supplement 3c*). Statistically, the number of MSHA pili shows a linear relationship with cell length (*Figure 1c*), indicating that the length density of MSHA pili is roughly constant for *V. cholerae*.

## MSHA pili mediate *V. cholerae* landing by acting as a brake and anchor

The MSHA pili, which are uniformly distributed across the cell surface, play a crucial role in the surface attachment of *V. cholerae* through pili-surface interactions (*Utada et al., 2014*). To elucidate the role of MSHA pili in the landing dynamics under viscoelastic conditions, we directly visualize the fluorescently labeled MSHA pili on *V. cholerae* swimming in a viscoelastic medium consisting of 2% Luria-Bertani (LB) and 1% methylcellulose (MC) (LB+MC).

Consistent with previous reports in normal aqueous solutions (*Utada et al., 2014*), the WT strain in LB+MC also exhibits orbiting behavior, characterized by multi-pass circular trajectories, and roaming behavior, characterized by highly exploratory, diffusive, trajectories. Typical roaming and orbiting trajectories in LB+MC are shown in *Figure 2* (see more examples in *Figure 2—figure supplement 1*). The roaming cell traces out a path that is linear over short distances, with a radius of gyration $R_g$=19.5 μm, and an average speed of 1.7 μm/s (see *Figure 2a,b*, *Video 1*). In contrast, the orbiting cell trajectory is much more circular with an average $R_g$=1.6 μm and an average speed of 1.1 μm/s (see *Figure 2c,d*, *Video 2*). A 3D plot of speed plotted along the trajectory in both examples show that both phenotypes make momentary pauses, where their speed slows down; this can be seen clearly in *Figure 2b*, where the cell motion near a surface displays a characteristic alternation between moving and stopping (*Figure 2b and d*).

Such pauses are suggested to be caused by MSHA pili-surface interactions (*Utada et al., 2014*). However, by recording fluorescence video sequences, we directly visualized the process, thereby providing direct evidence that the pauses are due to transient contact between MSHA pili and the surface. We show a transient pili-surface contact during orbiting in *Figure 2e*. In a time-lapse sequence, we show the stretching of a transiently attached pilus due to cell moving away from the

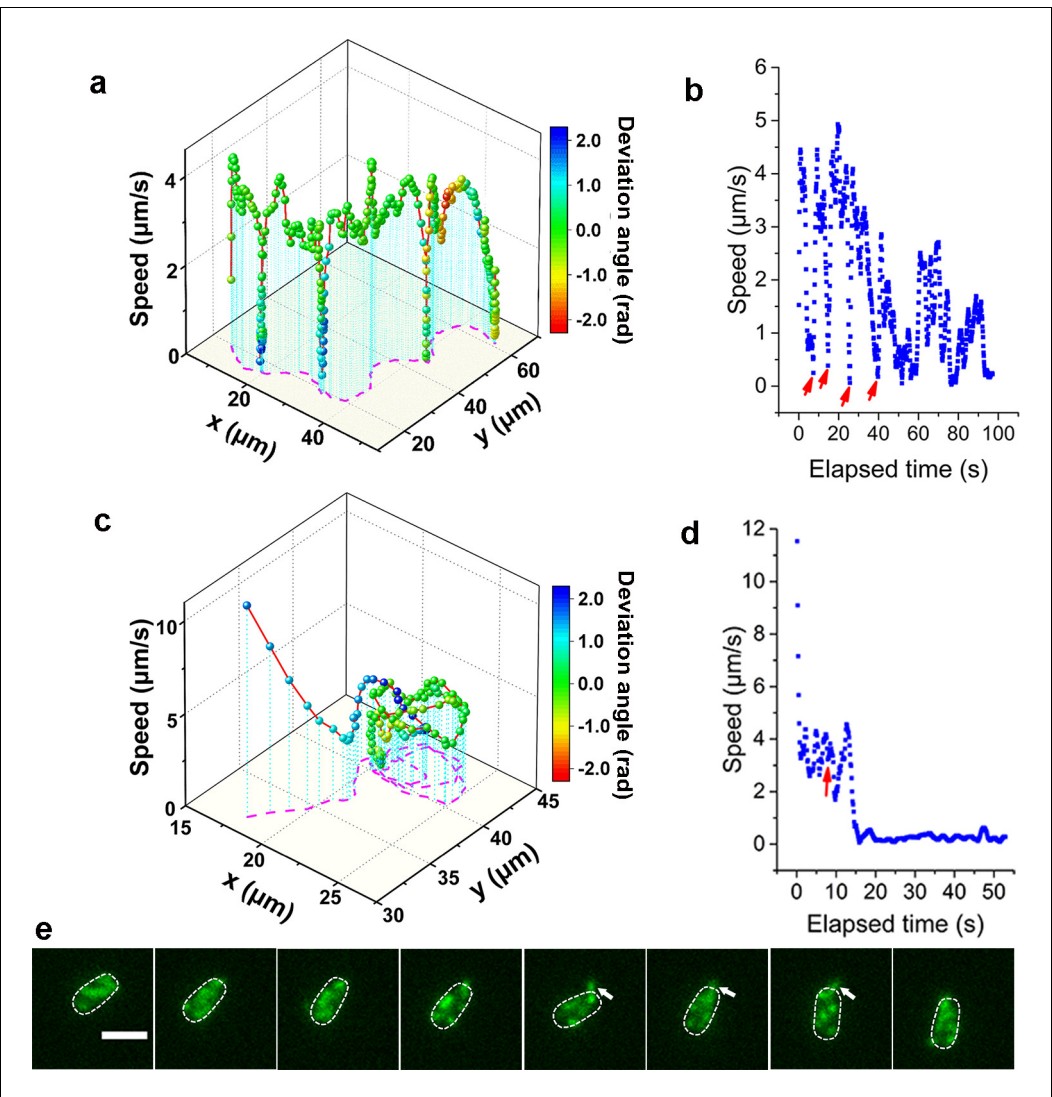

**Figure 2.** Analysis of roaming and orbiting, using cells of MSHA labeled MshAT70C. The 3D plot and speed changes over time of representative (**a, b**) roaming and (**c, d**) orbiting cells, respectively. The magenta dashed lines in panels (**a**) and (**c**) are the trajectories of cells and the color maps indicate the deviation angle. The arrows in panel (**b**) represent temporary attachment between MSHA pili and surface, where the speeds are close to 0. (**e**) Time-lapse images of the orbiting cell in panels (**c, d**) at 130 ms intervals (see **Video 2** for more details). The arrowheads show the stretched pilus, which corresponds to the red arrow in panel (**d**), indicating temporary attachment and stretching of pilus on the surface. Dashed lines indicate the estimated envelope of the cell body. Scale bar, 2 μm. MSHA, mannose-sensitive hemagglutinin.

The online version of this article includes the following source data and figure supplement(s) for figure 2:

**Source data 1.** Source data for *Figure 2a*.
**Source data 2.** Source data for *Figure 2b*.
**Source data 3.** Source data for *Figure 2c*.
**Source data 4.** Source data for *Figure 2d*.
**Figure supplement 1.** Quantitative analysis of roaming and orbiting by MSHA labeled MshAT70C in 2% LB with 1% MC.
**Figure supplement 2.** Switch of temporarily attached pili.

point of attachment. Subsequently, this pilus detaches from the surface as the cell continues to move, as indicated with the white arrowheads in *Figure 2e* (for more details, see *Video 2*). These results indicate that the MSHA pili can work as a brake to abruptly slow cell motion by transiently attaching to the surface. This is further confirmed by the observation that during the course of

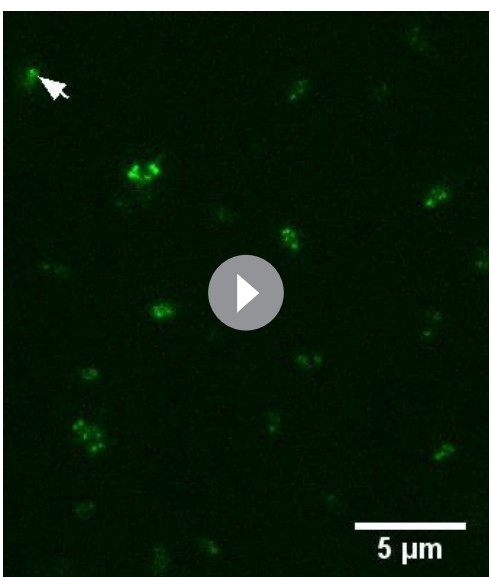

**Video 1.** Time-lapse fluorescence imaging showing a typical roaming cell (indicated by the arrowhead) with labeled MSHA pili in 2% LB+1% MC viscoelastic medium. This video was recorded every 390 ms for 98 s and displayed at 20 frames per second (fps).
https://elifesciences.org/articles/60655#video1

surface motion, different MSHA pili attach and detach, switching dynamically as the cell uses these as transient attachment points (*Figure 2—figure supplement 2* and *Video 3*). Such a switching of the specific MSHA pili that are engaging the surface is caused by the rotation of the cell body, which is required to balance the torque for flagellar rotation when cells swim. Thus, as the cell body rotates due to the rotation of the flagellar motor, different MSHA pili distributed on the cell body take turns approaching and receding from the surface. The switching of attached MSHA pili not only continues to slow down cell motion but also changes the direction of motion. Taken together, this indicates that the pili distribution on the cell body may also affect cell-surface interaction.

When the adhesion between MSHA pili and the surface is sufficiently strong, the attachment point can act as an anchor point. We demonstrate this by showing the deflection of the trajectory of a swimming cell by the attachment of a single anchoring MSHA pilus; here, linear motion is bent into a circular motion that is centered around the attachment point (see *Video 4*). We estimate the upper bound for the force exerted on the pilus during this motion by calculating the propulsive force of the flagellum using a resistive force theory model developed by *Lauga et al., 2006*. The force on the pilus will be less than this upper bound, partly due to the difference in direction between the pilus force and the propulsion force, which helps the cell rotate around the anchor point. Cell body rotations are an order of magnitude slower in the LB+MC solution used here than in water, so we estimate the flagellar rotation rate to be in the range of 10–100 rad/s. The viscosity of LB+MC is ~0.2 Pa·s and this gives us an upper bound for the force on the pilus of ~50 pN. In water, we estimate an upper bound of ~3 pN. These estimates are smaller than the 100 pN forces that pili can sustain (*Maier et al., 2002*). The anchoring of MSHA pilus eventually leads to the irreversible attachment of the cell.

## The landing sequence of *V. cholerae* includes three phases

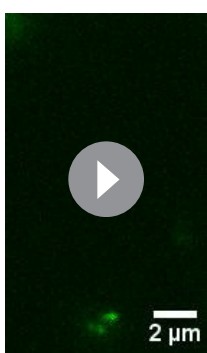

**Video 2.** Time-lapse fluorescence imaging showing a typical orbiting cell with labeled MSHA pili in 2% LB +1% MC medium. This video was recorded every 130 ms for 15 s and displayed at 10 fps.
https://elifesciences.org/articles/60655#video2

To further clarify the landing process, we labeled both flagellum and pili simultaneously using MshAT70CFlaAA106CS107C mutant. An example of the complete landing process of an orbiting cell is shown in *Figure 3*. Based on the

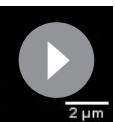

**Video 3.** Time-lapse fluorescence imaging showing switch of pili. When transient pauses happened, the attached pilus could be switched from one to another or more. See also *Figure 2—figure supplement 2*. This video was recorded every 70 ms for 10 s and displayed at 5 fps.
https://elifesciences.org/articles/60655#video3

pattern of motion displayed by the cell (*Figure 3a* and *Video 5*), we divide the landing process into three phases: running, lingering, and attaching. In the running phase (0–3.77 s), cells will swim and can perform roaming or orbiting. We note that misalignments between the flagellum and cell body axis tend to change the motion direction of the cell (*Figure 3a,b*). In the lingering phase (3.77–4.68 s), the cells demonstrate one of two states: a paused state or a tethered state, where the cell can move under the constraint of tethering pilus (see *Figure 3a* for the tethered state). At 3.77 s, one pilus attaches to the surface and acts as an anchor point to prevent the cell from moving away. Finally, in the attaching phase ($\geq$ 4.68 s), cells remain on the surface motionless during the observation period most likely since they have effected irreversible attachment. Upon irreversible cell attachment, some of the free MSHA pili become attached to the surface firmly while others demonstrate fluctuations punctuated with intermittent attachment to the surface (*Video 6*).

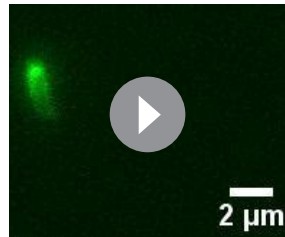

**Video 4.** Time-lapse fluorescence imaging showing linear motion bent into a circular motion that is centered around the attachment point between MSHA pili and the surface, which can act as an anchor point. This video was recorded every 130 ms for 8 s and displayed at 10 fps.

https://elifesciences.org/articles/60655#video4

During cell landing, transitions between the running and lingering phases, as well as between the two states of the lingering phase are observed, respectively. The measured conditional probabilities $q_{ij}$ that cell transitions from state $i$ to $j$ show that the running phase has a relatively lower $q_{rt}$ to the tethered state (~22%) but a higher $q_{rp}$ to the paused state (~78%). Similarly, the paused state has a higher $q_{pr}$ than $q_{pt}$. In contrast, the tethered state shows similar $q_{tr}$ and $q_{tp}$, which are 45% and 55%, respectively (*Figure 3c*).

The single-cell dynamics in each specific phase/state is also characterized quantitatively. In the running phase of *V. cholerae*, we found that the period for body rotation is generally distributed between 0.25 and 2 s and is centered at ~0.7 s (the rotation rate was ~1.5 Hz) in LB+MC (*Figure 4a*). We measured the swimming speed, *v*, and the cell-body rotation rate, $\omega_c$, for each cell, and plotted *v* as a function of $\omega_c$ (see *Figure 4b*). By fitting the data, we found that *v* linearly increases with $\omega_c$ with a slope of $|v/\omega_c|$=2.48±0.04 µm/radian.

By contrast, a cell in the tethered state typically performs a circular motion around the attachment point (red dots in *Figure 4c*). The direction of the circular motion is also dynamic and can switch from counterclockwise (CCW) to clockwise (CW) presumably due to a switch in the rotation direction of the flagellar motor (see 2.6 s, *Figure 4c*). Angular velocity is roughly constant during each circular-motion interval (i.e., in each CCW or CW period) and quickly changes sign after CCW-CW switching (*Figure 4d* and *Video 7*). Due to the distribution of pili across the cell body, tethering can occur at a pole or under the body, which leads to cells standing vertically or lying down horizontally to the surface, respectively. We find that standing tethered cells perform a faster circular motion (mean angular speed=8.5±1.9 rad/s) than lying ones (mean angular speed=3.0±2.1 rad/s) (*Figure 4e*). For the horizontal cells, different MSHA pili may be used to further anchor the cell to the surface. For example, two horizontally tethered cells demonstrate different tethered-motion trajectories depending on the location of the anchoring MSHA pilus (*Figure 4—figure supplement 1*). In addition to the fact that unattached pili may increase the likelihood that the cell will make the irreversible attachment, we observe that MSHA pili appear to be able to attach to the surface along their entire length, and not just the tip (*Video 8*).

Interestingly, we find that the flagellum of attached cells frequently continues to rotate (*Video 5*, after 4.68 s), indicating that even after cell attachment, the flagellar motor is still active for some period. The flagellum will eventually stop rotating after a cell stays long enough on the surface (*Video 9*).

## Role of MSHA pili in cell landing is more apparent in viscoelastic (non-Newtonian) fluids than in viscous Newtonian fluids

To further investigate the dependence of MSHA pili function and hence cell landing on viscoelasticity, we compared cell motion behavior (*Figure 5* and *Figure 5—figure supplement 1*) obtained in

2% LB, which is a Newtonian fluid with a viscosity ~1 cP at 30°C (*Utada et al., 2014*) and in 2% LB +1% MC, which is a non-Newtonian fluid and has a shear-dependent viscosity (*Figure 6*), for both WT and △*mshA* cells.

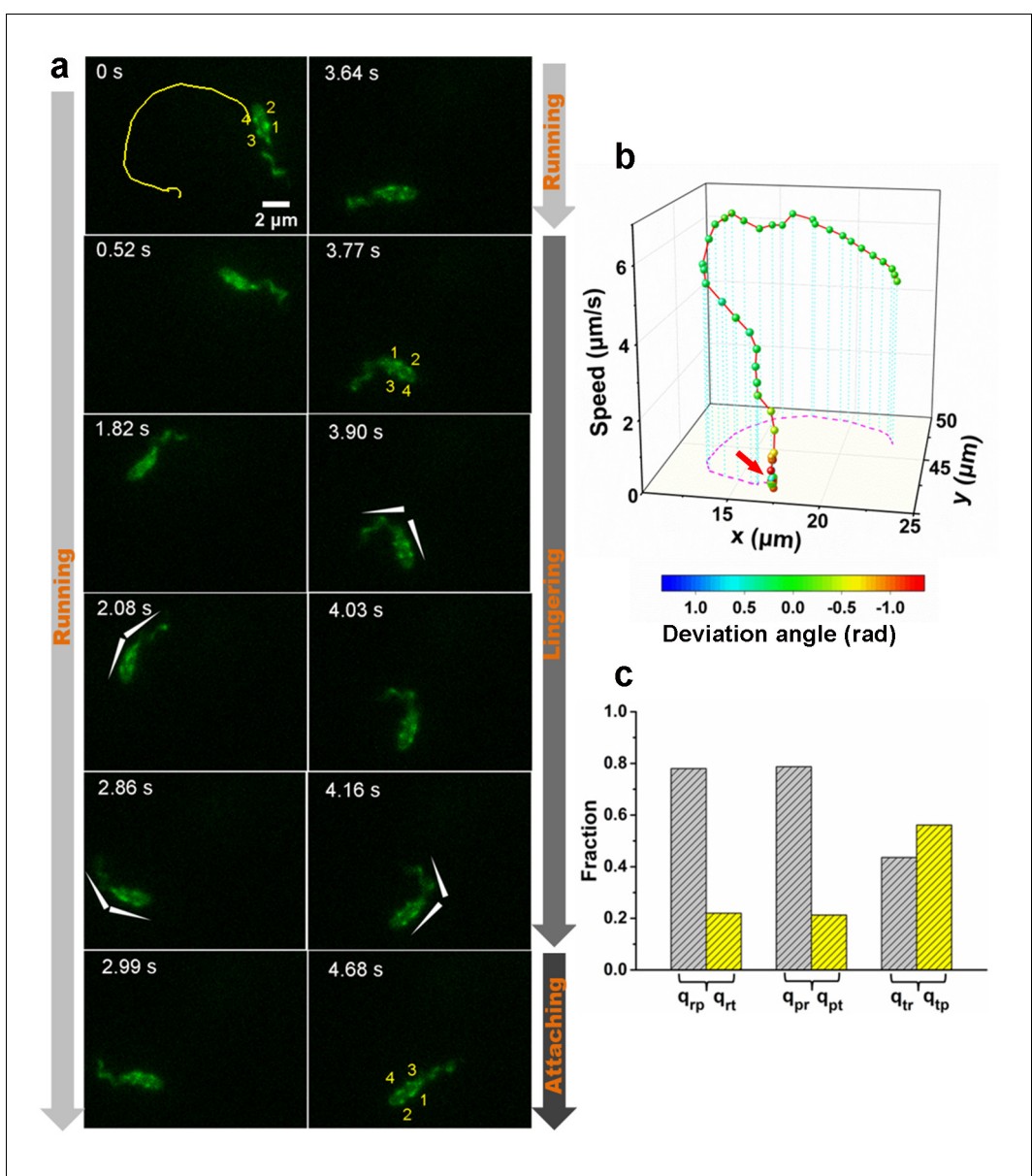

**Figure 3.** An example of a typical landing sequence of a *Vibrio cholerae* cell with MSHA pili and flagella both labeled (MshAT70CFlaAA106CS107C). (**a**) Representative image sequences showing the behavior of MSHA pili and flagellum. For easy identification, four pili of the example cell in (**a**) are numbered from 1 to 4, which revolve around the major axis of the cell periodically as the cell swims. The white arrowheads indicate the orientation of the cell body and flagellum. (**b**) A 3D plot of speed and deviation angle of the representative cell in panel (**a**) over its trajectory. The red arrow in panel (**b**) represents the position, where the pili touch surface, causing a deflection. (**c**) The conditional probabilities $q_{ij}$ that the bacterium transitions from state $i$ to $j$. The number of transition events used for estimating these conditional probabilities is 666. r: running state, t: tethered state, p: paused state. MSHA, mannose-sensitive hemagglutinin.

The online version of this article includes the following source data for figure 3:

**Source data 1.** Source data for *Figure 3b*.
**Source data 2.** Source data for *Figure 3c*.

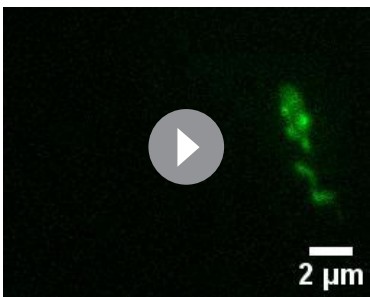

**Video 5.** Time-lapse fluorescence imaging showing dynamic movement of a MshAT70CFlaAA106CS107C cell with the labeled flagellum and MSHA pili in 2% LB +1% MC medium. This video was recorded every 130 ms for 13 s and displayed at 10 fps.
https://elifesciences.org/articles/60655#video5

**Video 6.** Time-lapse fluorescence imaging showing five MSHA pili of a WT cell stuck to the surface and kept still or fluctuated frequently. This video was recorded every 460 ms for 25 s and displayed at 10 fps.
https://elifesciences.org/articles/60655#video6

We observed WT cells to demonstrate roaming and orbiting motilities in both solutions (*Figure 5a*). The histograms of deviation angle of each type of motility obtained in the two solutions are also similar (*Figure 5—figure supplement 1a and b*). These results indicate that the roaming and orbiting motilities of cells are robust against the tested viscoelastic environment. Although the general motility pattern is similar in both solutions, the motion of cells, as expected, is slowed significantly in LB+MC. The average speed of WT cells for near-surface motion is reduced by ~22 times from 86.7±32.9 μm/s (mean ± standard deviation) in 2% LB to 3.8±2.6 μm/s in LB+MC. Similarly, the average speed of △*mshA* cells is also decreased by ~12 times from 80.0±15.0 μm/s in 2% LB to 6.5±1.4 μm/s in LB+MC. The slowing of the motion can also be seen clearly from their mean square displacement curves (*Figure 5—figure supplement 1c,d*), which have similar shapes but very different time scales.

However, WT and △*mshA* cells also show differences in their motility behavior in these two solutions (*Figure 5*). In LB+MC, WT cells tend to land on the surface soon after approaching it (less than one round in orbiting motility) and more tethered motion is observed, which leads to more irregular and tortuous trajectories and smaller $R_g$ for WT cells compared with the case of 2% LB (*Figure 5c*). By contrast, △*mshA* cells show very similar $R_g$ distributions in the two types of solutions (*Figure 5c*). More interestingly, compared with WT, in LB+MC, a large proportion of △*mshA* cells show orbiting for a substantial number of cycles, as shown in *Figure 5b*. Quantitatively, this can be seen in the calculated mean path length, $\bar{l}$, which is 39.7±51.2 μm for WT and 72.5±99.1 μm for △*mshA* in LB+MC, whereas the corresponding value in 2% LB is 58.7±63.1 μm for WT and 47.2±50.8 μm for △*mshA*. To see how the role of MSHA pili varies with viscoelasticity, we calculated the ratio of the respective mean path lengths of △*mshA* and WT, $\bar{l}_{\Delta mshA}/\bar{l}_{WT}$, for each type of solution; this gives ~1.8 in LB+MC and ~0.8 in 2% LB, respectively (*Figure 5d and e*). This shows that the loss of MSHA pili results in a significant increase in mean path length in LB+MC relative to 2% LB. Moreover, such prolonged orbiting motions of △*mshA* were not observed in a 20% Ficoll solution (*Figure 5*), which has a high viscosity but still belongs to the class of Newtonian fluids (*Winet, 1976*). As shown in *Figure 5*, the cell motility behaviors in 20% Ficoll are similar to those in 2% LB only except that the average cell speed for near-surface motion is dramatically lower in 20% Ficoll, which is 9.3±4.3 μm/s for WT and 10.0±3.5 μm/s for △*mshA*. The cell trajectories in 20% Ficoll are similar to those in 2% LB only and consequently, the ratio of $\bar{l}_{\Delta mshA}/\bar{l}_{WT}$ is ~0.75 in 20% Ficoll, very close to ~0.8 in 2% LB only. Taken together, these results indicate that the elastic properties of viscoelastic solutions can also affect cell motility behavior and the role of MSHA pili as a braking and anchoring machine in cell landing is more apparent in viscoelastic (non-Newtonian) fluids than in viscous Newtonian fluids.

## Discussion

The first step in *V. cholerae* biofilm formation is the transition from planktonic swimmers to the stationary surface-attached cells; this process is mediated by the landing process (*Teschler et al., 2015*). In this study, the combination of cell appendage labeling and high-resolution spatio-temporal imaging allows us to quantitatively deconstruct the landing process into three

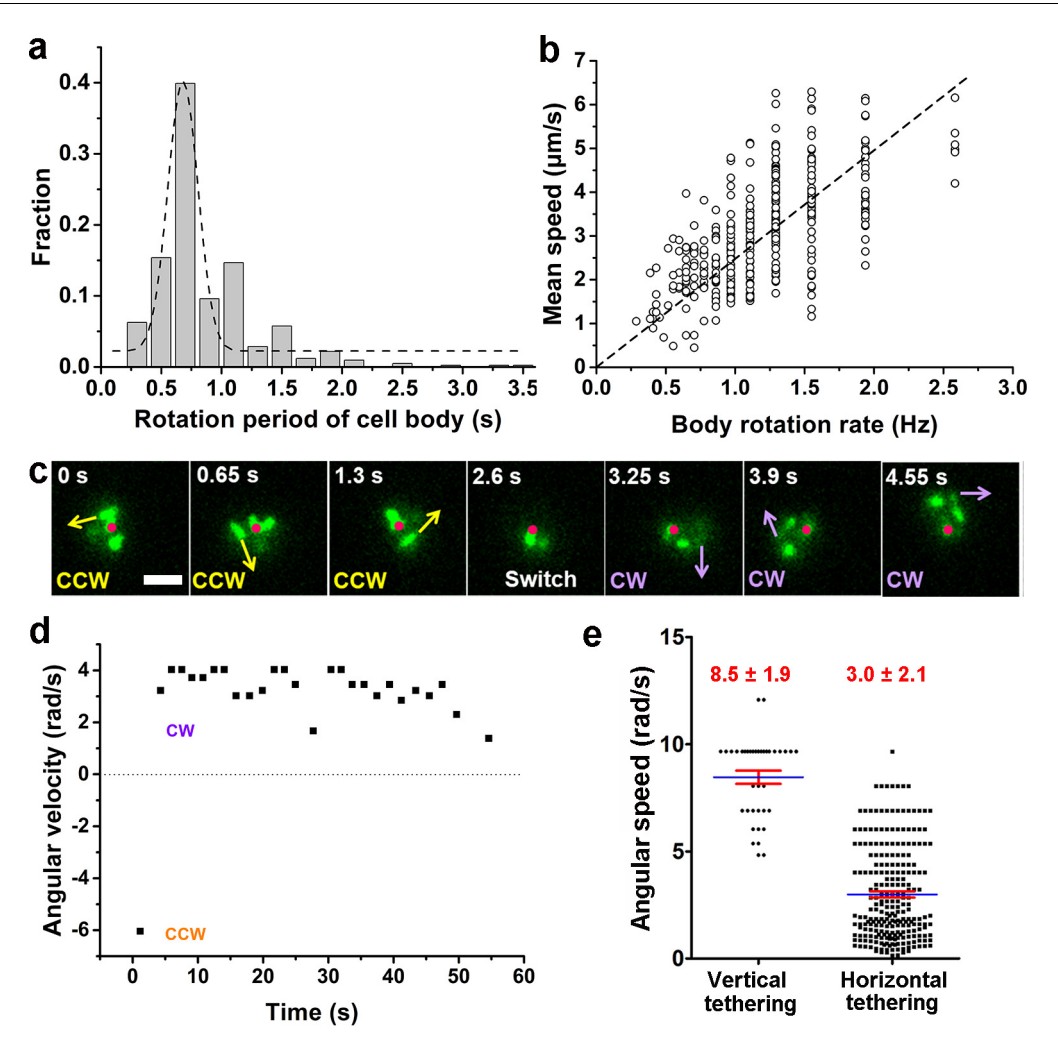

**Figure 4.** Characterization of running and tethered cells. (**a**) Distribution of the rotation period of the cell body. The dashed line represents Gaussian fitting. A total of 416 rotation events from 54 cells were used for statistical analysis. (**b**) Measured relation between the rotation rate of the cell body and the mean swimming speed of the cell. The dotted line represents linear fitting result. $N_{cell}$=47. (**c**) An example of a typical tethered motion, showing a cell performing a circular motion around a center point (the red dot) with the direction of motion (noted by arrows) switched from CCW to CW. Scale bar, 2 μm. (**d**) The angular velocity of the tethered cell in panel (**c**) over a short duration showing a pair of CW (positive angular velocity) and CCW (negative angular velocity) intervals; (**e**) Distribution of angular speed of circular motion for horizontal (241 intervals from 25 cells) and vertical (38 intervals from 5 cells) tethered cells. CCW, counterclockwise; CW, clockwise.

The online version of this article includes the following source data and figure supplement(s) for figure 4:

**Source data 1.** Source data for *Figure 4a*.
**Source data 2.** Source data for *Figure 4b*.
**Source data 3.** Source data for *Figure 4d*.
**Source data 4.** Source data for *Figure 4e*.
**Figure supplement 1.** Examples show positions of two poles and centroid of tethered motility.

stages: running, lingering, and attaching, as summarized in *Figure 7*. During the running phase, cell motion is powered by flagellar rotation, which simultaneously induces a counter-rotation of the cell body. When swimming cells come to within a distance that is comparable to the length of a typical pilus from a surface, dangling pili may brush against the surface, thereby deflecting the trajectory. Typical MSHA pili are ~0.4–1.2 μm in length. During near-surface swimming, cell body rotation actively brings MSHA pili into close proximity with the underlying surface where friction between pili

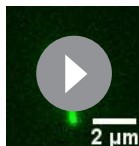

**Video 7.** Time-lapse fluorescence imaging showing a typical tethered cell performing a circular motion around a fixed point with the direction of motion switched from CCW to CW. See also *Figure 4c*. This video was recorded every 130 ms for 6 s and displayed at 5 fps.

https://elifesciences.org/articles/60655#video7

and the surface can slow the cells, or, transient adhesions can be made, which may even arrest cell motion. Here, we make an analogy to the slow down and stop affected by the brake system of a car. During near-surface swimming, it has been suggested that hydrodynamic forces cause the cell bodies of swimming rod-like bacteria to take on a tilted, non-parallel, orientation to the surface (*Vigeant et al., 2002*). In the case of *P. aeruginosa*, whose TFP are distributed with a strong bias toward a particular pole (*Skerker and Berg, 2001*), pili-surface contact will depend on which pole is closer to the surface. In contrast, the homogeneous distribution of MSHA pili on *V. cholerae* (see *Figure 1c*) may be more efficient at slowing such tilted cell bodies by increasing the probability that pili encounter the surface relative to bacteria with biased pili distributions.

If the contact-induced adhesion between MSHA pili and the surface is sufficiently strong to arrest forward motion, the cell will either pause or commence tethered motion centered about the point-of-adhesion. Cells rotating at an angle closer to the surface have a slower angular velocity (*Figure 4e*), to which hydrodynamic effects presumably have an important contribution (*Bennett et al., 2016*). This suggests that for cells demonstrating tethered motion, a progressive twisting of the surface-attached pilus fiber during the circular motion of cells may gradually cause the circular motion to stop by pulling the cell body ever closer to the surface. Although twitching has not to be observed in *V. cholerae*, this is one mechanism by which retraction-like dynamics may be achieved (*Charles et al., 2019*), possibly in tandem with actual retraction of MSHA pili, which has been shown recently in a different strain of *V. cholera* (*Floyd et al., 2020*). Under our conditions, we have not observed MSHA pili retraction events nor have we seen bacterial cells that gradually acquire fluorescence when only maleimide dyes were used. These results are consistent since in bacteria where pilus retraction does occur, such as in the TAD pili of *C. crescentus* (*Ellison et al., 2017*), ChiRP of *V. cholera* (*Ellison et al., 2018*), and TFP of *P. aeruginosa* (*Skerker and Berg, 2001*), the cell body gradually becomes fluorescent due to internalization of labeled pilin by retraction. Such phenotypical differences may be due to the different experimental conditions used in each study and require more work to fully elucidate.

In addition to possible hydrodynamical effects, our observation that MSHA pili are able to adhere to surfaces along their entire length implies that cells can enhance their chances of attachment through the possibility of increased adhesion between the surface and the whole pilus filament. The ability to adhere not only at the distal tip, contrasts with the TFP of *P. aeruginosa* (*Skerker and Berg, 2001*) and ChiRP of *V. cholerae* (*Ellison et al., 2018*), which show that the pilus-subject interactions are mainly mediated by the pilus tip. Thus, for *V. cholerae*, the strength of adhesion between a cell and a surface that is mediated by an individual MSHA pilus appears to be more complicated to model than with a single point of attachment. Rather, cells can enhance the adhesion strength by increasing both the segment length and the number of MSHA pili adhered to the surface. These two effects will facilitate irreversible attachment in *V.*

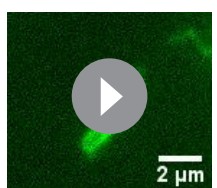

**Video 8.** Time-lapse fluorescence imaging showing different adhesion points of a pilus. When the tip of the pilus was free (~3.5 s), the upper part of the pilus was still capable of keeping the cell adhered. This video was recorded every 130 ms for 13 s and displayed at 10 fps.

https://elifesciences.org/articles/60655#video8

**Video 9.** Time-lapse fluorescence imaging showing the motion evolution of the flagellum from rotating to stopping eventually. This video was recorded every 130 ms for 10 s and displayed at 10 fps.

https://elifesciences.org/articles/60655#video9

*cholerae.*

Similar running and lingering phases in near-surface motion have also been reported in entero-haemorrhagic *E. coli* (EHEC) cells (*Perez Ipiña et al., 2019*), where the results suggest that by choosing the optimal transition rates, EHEC bacterial diffusivity is maximized and surface exploration efficiency is greatly improved. In future work, it will be interesting to apply a similar analysis in *V. cholerae.*

In this study, the data collection of *V. cholerae* cells was performed mainly in a 2% LB medium supplemented with 1% MC. Rheological measurements show that its viscosity is shear dependent and has a higher storage modulus than loss modulus in the tested oscillation frequency range; thus, it is a non-Newtonian fluid (*Figure 6a–c*). On the other hand, the rheological measurements of an 18.7% (w/w) mucin solution in 2% LB, which was obtained by measuring the concentration of mixtures scraped from the fresh mouse intestine surfaces, show similar viscoelastic behavior with a higher storage modulus than loss modulus in the same tested oscillation frequency range (*Figure 6d–f*). Therefore, compared with Ficoll solutions, which are Newtonian fluids, LB+MC better simulates the viscoelastic environment that *V. cholerae* cells encounter in the mucus layer of animal intestines. In such viscoelastic environments, *Millet et al., 2014* observed considerable differences of bacterial localization in different parts of the small intestine and found that *V. cholerae* motility exhibits a regiospecific influence on colonization, indicating that viscoelastic intestinal mucin is a key factor limiting colonization. However, it is technically challenging to directly observe cell motion in mucus solutions made from lyophilized mucus powders or fresh mucus-containing solutions scraped from the surfaces of mouse intestines due to the complicated inhomogeneous environment with too many impurities (data not shown). In this work, by direct visualization of pili and flagellum of cells during their landing process in LB+MC that rheologically mimics actual mucus solutions, we find that *V. cholerae* cells are able to move well in this viscoelastic solution under the conditions tested. Moreover, we show that the effect of MSHA pili as a braking and anchoring machine on cell landing is more apparent in LB+MC than in either 2% LB or 20% Ficoll solutions, suggesting that MSHA pili may play an even more important role for surface attachment in viscoelastic, non-Newtonian, environments such as in the mucus layer of small intestines.

To summarize, in this work, using fluorescence imaging with labeled pili and flagellum, we show a comprehensive picture of the landing dynamics of *V. cholerae* cells in viscoelastic environments and provide direct observational evidence of the role of MSHA pili during cell landing. Our work provides fundamental insight into the mechanism of *V. cholerae* surface attachment and we hope that it may lead to methods that prevent and control *V. cholerae* infection.

## Materials and methods

**Key resources table**

| Reagent type (species) or resource | Designation | Source or reference | Identifiers | Additional information |
|---|---|---|---|---|
| Parent strain (*Vibrio cholerae*) | C6706 | *Joelsson et al., 2006* | | |
| Plasmids (DH5α λpir) | pWM91 | *Metcalf et al., 1996* | | Suicide vector |
| Chemical compound, drug | Alexa Fluor 488 C5 Maleimide | Thermo Fisher Scientific | Cat. #: A10254 | |
| Chemical compound, drug | Alexa Fluor 546 C5 Maleimide | Thermo Fisher Scientific | Cat. #: A10258 | |
| Chemical compound, drug | Methyl cellulose (MC) | Solarbio | Cat. #: M8070 | |
| Chemical compound, drug | Ficoll 400 | Yuanye Bio-Technology | Cat. #: 26873-85-8 | MW=400 kDa |
| Software | GraphPad Prism software | | RRID:SCR_002798 | |
| Software | Matlab | | RRID:SCR_001622 | matlab R2015a |
| Software | Leica LAS-X | | RRID:SCR_013673 | |

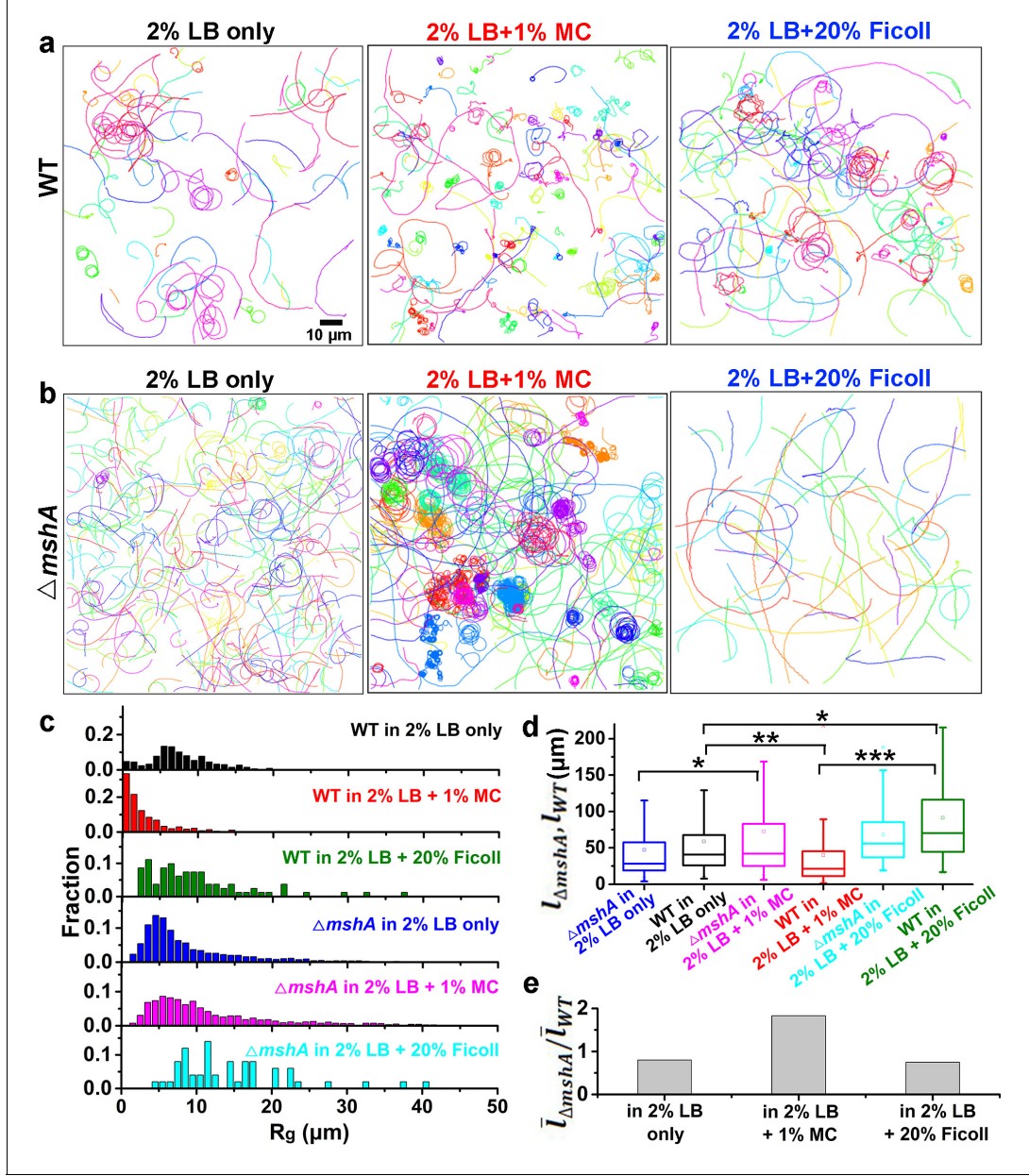

**Figure 5.** Role of MSHA pili in cell landing is more apparent in viscoelastic non-Newtonian solutions than viscous Newtonian fluids. (**a**) Examples of WT cell trajectories showing both roaming and orbiting motilities in 2% LB only, 2% LB with 1% MC, and 2% LB with 20% Ficoll; (**b**) Examples of cell trajectories of △*mshA*; (**c**) Histograms of $R_g$ of WT and △*mshA* in different solutions; (**d**) A box plot summary of path lengths of WT and △*mshA*. Statistical significance was determined with one-way ANOVA followed by Tukey's multiple comparison test comparing the different groups (*$p<0.05$; **$p<0.01$; ***$p<0.001$). The data were analyzed using the Prism 5.0 software program (GraphPad Software, La Jolla, CA, USA). (**e**) The ratio of mean path length between △*mshA* and WT, $\bar{l}_{\triangle mshA}/\bar{l}_{WT}$. LB, Luria-Bertani; MC, methylcellulose; MSHA, mannose-sensitive hemagglutinin.

The online version of this article includes the following source data and figure supplement(s) for figure 5:

**Source data 1.** Source data for *Figure 5c*.

**Source data 2.** Source data for *Figure 5d*.

**Source data 3.** Source data for *Figure 5e*.

**Figure supplement 1.** Motility characterization of WT and △*mshA* cells in 2% LB only and in 2% LB with 1% MC.

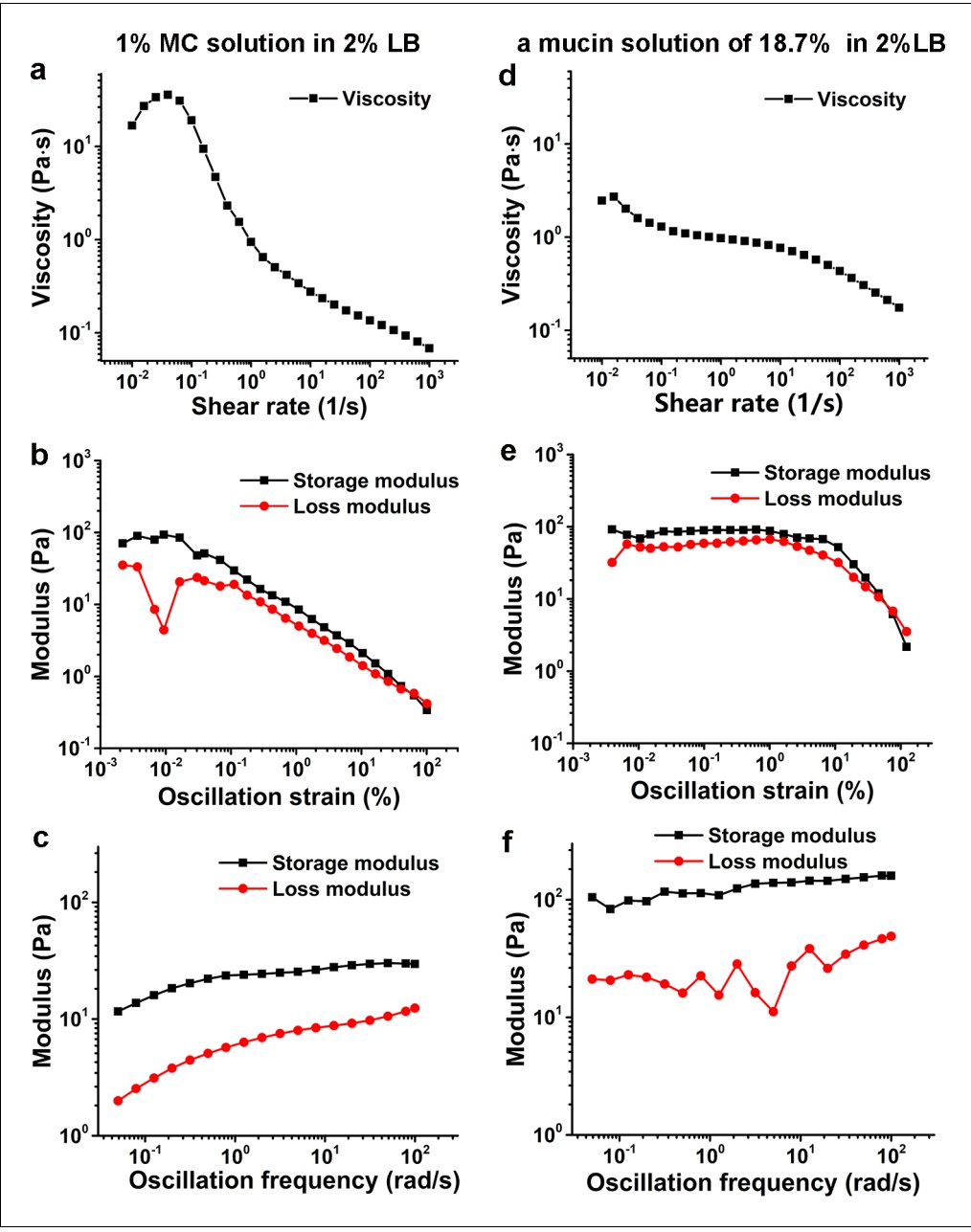

**Figure 6.** Characterization of viscoelasticity of the 1% MC solution in 2% LB (a–c) and a mucin solution of 18.7% (w/w) in 2% LB (d–f) at 26°C. (a) and (d) show viscosity as a function of shear rate. (b) and (e) display modulus as a function of the oscillation strain, using cone-plate geometry. (c) and (f) show modulus as a function of the oscillation frequency under an oscillation strain of 0.1%, using cone-plate geometry. LB, Luria-Bertani ; MC, methylcellulose.

The online version of this article includes the following source data for figure 6:

**Source data 1.** Source data for *Figure 6a*.
**Source data 2.** Source data for *Figure 6b*.
**Source data 3.** Source data for *Figure 6c*.
**Source data 4.** Source data for *Figure 6d*.
**Source data 5.** Source data for *Figure 6e*.
**Source data 6.** Source data for *Figure 6f*.

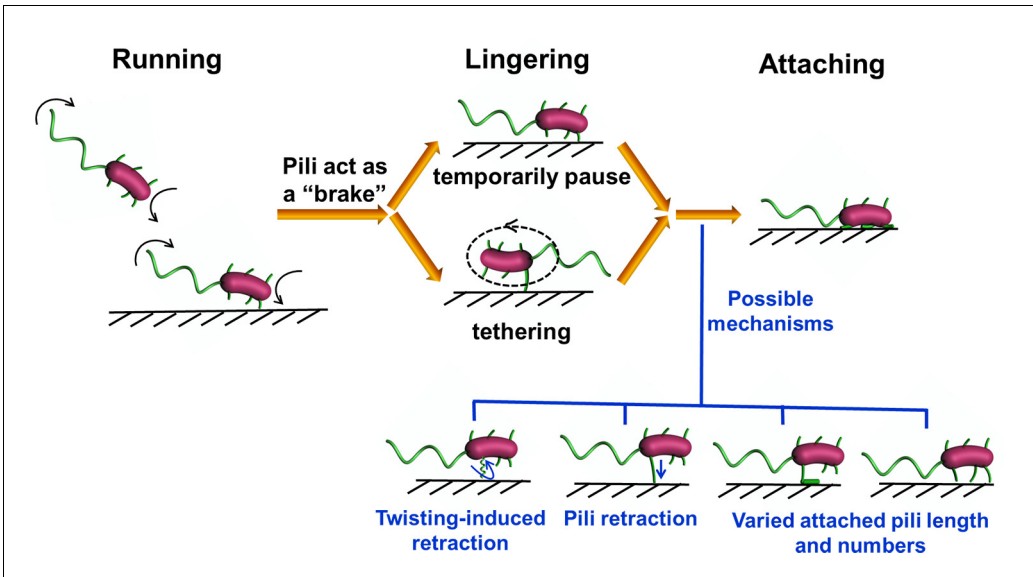

**Figure 7.** Diagrammatic sketch for the landing process of *Vibrio. cholera* cells on a substrate.

## Bacterial strains

Bacterial strains, plasmids, and primers used in this study are listed in *Table 1* and *Supplementary file 1*, Table S1. *V. cholerae* El Tor C6706 (*Joelsson et al., 2006*) was used as a parental strain in this study. C6706 and mutants were grown at 30°C or 37°C in LB supplemented with 100 μg/mL streptomycin, 50 μg/mL kanamycin, 1 μg/mL chloromycetin where appropriate. *E. coli* strains harboring plasmids were grown at 37°C in LB supplemented with 100 μg/mL ampicillin. The optical densities of bacterial cultures were measured at 600 nm ($OD_{600}$) using a UV-vis spectrophotometer.

## Flagellin and pilin mutagenesis

Following the protocol in *Ellison et al., 2019*; *Ellison et al., 2017*, we first predicted 10 amino acid residues in *V. cholerae* flagellin FlaA for cysteine replacement. Then, the *flaA* knockout and FlaA sequences containing the FlaAA106C, FlaAS107C, FlaAA106CS107C, FlaAE332C, FlaAG23C, FlaAN26C, FlaAN83C, FlaAS325C, FlaAS87C, FlaAS376C, and FlaAV117C knock-in were constructed using the MuGENT method (*Dalia et al., 2014*). The FlaAA106C, FlaAS107C, and FlaAA106CS107C knock-in were constructed by cloning the fragment into the suicide vector pWM91 containing a *sacB* counter-selectable marker (*Metcalf et al., 1996*). The plasmids were introduced into *V. cholerae* by conjugation and mutations were selected for double homologous recombination events. The MshAT70C mutation that can be successfully labeled with thiol-reactive maleimide dyes has been described previously (*Ellison et al., 2017*), and MshAT70C was constructed using the MuGENT method to light MSHA pilus. All mutants were confirmed by DNA sequencing.

## Hemagglutination assays

MSHA by *V. cholerae* was measured as described previously (*Gardel and Mekalanos, 1996*). Briefly, bacteria were grown to the mid-logarithmic phase in the LB medium. Initial concentrations of approximately $10^{10}$ CFU/mL were twofold diluted with Krebbs-Ringer-Tris (KRT) buffer in U-bottomed wells of 96-sample microtiter dishes. Sheep erythrocytes were washed in phosphate-buffered saline (PBS) and resuspended in KRT buffer for a final concentration of 10% (v/v). Equivoluminal erythrocyte was added into serially diluted bacterial suspensions and the plates were gently agitated at room temperature (RT) for 1 min. Samples were checked for hemagglutination after 2 hr at RT.

**Table 1.** Strains used in this study.

| Strain | Description | Source or reference |
|---|---|---|
| Parent strain (*Vibrio cholerae*) | C6706 Sm$^R$ | *Joelsson et al., 2006* |
| Δ*mshA* | C6706 Sm$^R$, VC1807:: Cm$^R$, *mshA* knockout | This study |
| Δ*flaA* | C6706 Sm$^R$, VC1807:: Cm$^R$, *flaA* knockout | This study |
| MshA$^{T70C}$ | C6706 Sm$^R$, VC1807:: Km$^R$, MshAT70C | *Ellison et al., 2017* |
| FlaA$^{A106C}$ | C6706 Sm$^R$, VC1807:: Cm$^R$, FlaAA106C | This study |
| FlaA$^{S107C}$ | C6706 Sm$^R$, VC1807:: Cm$^R$, FlaAS107C | This study |
| FlaA$^{A106CS107C}$ | C6706 Sm$^R$, VC1807:: Cm$^R$, FlaAA106CS107C | This study |
| FlaA$^{E332C}$ | C6706 Sm$^R$, VC1807:: Cm$^R$, FlaAE332C | This study |
| FlaA$^{G23C}$ | C6706 Sm$^R$, VC1807:: Cm$^R$, FlaAG23C | This study |
| FlaA$^{N26C}$ | C6706 Sm$^R$, VC1807:: Cm$^R$, FlaAN26C | This study |
| FlaA$^{N83C}$ | C6706 Sm$^R$, VC1807:: Cm$^R$, FlaAN83C | This study |
| FlaA$^{S325C}$ | C6706 Sm$^R$, VC1807:: Cm$^R$, FlaAS325C | This study |
| FlaA$^{S87C}$ | C6706 Sm$^R$, VC1807:: Cm$^R$, FlaAS87C | This study |
| FlaA$^{S376C}$ | C6706 Sm$^R$, VC1807:: Cm$^R$, FlaAS376C | This study |
| FlaA$^{V117C}$ | C6706 Sm$^R$, VC1807:: Cm$^R$, FlaAV117C | This study |
| MshA$^{T70C}$, Δ*flaA* | C6706 Sm$^R$, VC1807:: Km$^R$, *flaA* knockout | This study |
| MshA$^{T70C}$, FlaA$^{A106C}$ | C6706 Sm$^R$, VC1807:: Km$^R$, FlaAA106C | This study |
| MshA$^{T70C}$, FlaA$^{S107C}$ | C6706 Sm$^R$, VC1807:: Km$^R$, FlaAS107C | This study |
| MshA$^{T70C}$, FlaA$^{A106CS107C}$ | C6706 Sm$^R$, VC1807:: Km$^R$, FlaAA106CS107C | This study |

The results of the hemagglutination assay test show that MshAT70C displays similar behavior to WT, which indicates that the point mutation in MSHA does not affect MSHA pilus function (*Figure 1—figure supplement 4*).

## Preparation of viscous solutions

To change the solution viscosity, MC (M20, 4000 cp, Solarbio, China) solutions were prepared by dissolving 0% and 1% (w/v) MC in 2% LB motility medium (containing 171 mM NaCl). 20% (w/v) Ficoll 400 (MW=400 kDa, Yuanye Bio-Technology Co., Ltd, China) dissolved in 2% LB medium was also prepared for a control experiment.

## Preparation of the small intestinal mucus samples from ICR female mice

Six-week-old female ICR mice were provided with drinking water with 10 g/L streptomycin and 0.2 g/L aspartame for 2 days. To clean the small intestine content, food was removed 24 hr prior to the start of the experiments. The mice were killed and the small intestine was cut open with sterile scissors and the mucus layer was gently extracted with a cell scraper. Samples were weighed to obtain the wet weight first and then frozen overnight at −80℃. After that, they were freeze dried using a freeze dryer (Labconco, USA) in −40℃ low-temperature vacuum. Next, these dried samples were weighed again to obtain the dry weight. The average value of the dry/wet ratio from three repeats is 18.7%.

## Cell imaging

For the *V. cholerae* motility observation in 2% LB without MC, overnight cultures in LB were resuspended and diluted with 2% LB to an $OD_{600}$ ranging from 0.01 to 0.03. Then, the bacterial suspension was injected into a flow cell, which contained the same media. Imaging was performed using a Phantom V2512 high-speed camera (Vision Research, USA) collecting ~200,000 bright-field images at 5 ms resolution with a 100× oil objective on a Leica DMi8 inverted microscope (Leica, Germany) at a set temperature value of 30℃.

For the *V. cholerae* motility observation in 2% LB with 1% MC (henceforth, this medium is referred to LB+MC), overnight cultures in LB were resuspended and diluted with LB+MC to a final $OD_{600}$ of 0.01–0.03. Then, the bacteria were incubated at 37℃ for 20 min to allow them to adapt to the new environment and were then used immediately. Bacteria samples were pipetted onto standard microscope slides with an 8 mm diameter spot and then were sealed with a coverslip using a 1 mm thick secure spacer. Imaging was performed using an EMCCD camera (Andor iXon Ultra 888) collecting ~10,000 bright-field images with a time resolution of 90 ms at a set temperature value of 30℃. Similar protocols were carried out for observations in 20% Ficoll solutions and in the small intestinal mucus samples of mice.

## Cell-tracking algorithms and analysis

The images were preprocessed using a combination of software and algorithms adapted from the methods described (*Lee et al., 2016*; *Utada et al., 2014*; *Zhao et al., 2013*) and written in MATLAB R2015a (Mathworks) by subtracting the background, scaling, smoothing, and thresholding. After image processing in this way, the bacteria appear as bright regions. The bacteria shape was fit with a spherocylinder. Then, the geometric information of the cell, such as the location of the centroid and two poles, and the length and width of the bacterium were collected. Trajectory reconstruction was also achieved for further analysis.

The motility parameters (*Utada et al., 2014*), such as instantaneous speed, deviation angle, radius of gyration ($R_g$), MSD, and mean path length $\bar{l}$ were calculated to further characterize the near-surface motility of *V. cholerae*. The instantaneous speed was calculated via $|r_{i+1}−r_i|/\triangle t$, where $r_i$ is the cell position vector in frame $i$ and $\triangle t$ is the time interval between two consecutive frames. The deviation angle of cell motion is defined as the angle between its cell body axis and the direction of motion. The radius of gyration, $R_g$, is a statistical measure of the spatial extent of the domain of motion given by an ensemble of points that define a trajectory (*Rubenstein, 2003*). The square of this quantity is defined as $R_g^2 = \frac{1}{N}\sum_{i=1}^{N}\left(\vec{R}_i - \vec{R}_{cm}\right)^2$, where $N$ is the number of points in the tracked trajectory, $\vec{R}_i$ is the position vector corresponding to the $i$th point on the trajectory, $\vec{R}_{cm}$ is the position vector of the center-of-mass. The MSD of cells was calculated via $\left\langle \Delta r^2(\tau) \right\rangle = \left\langle \left[r(t + \tau) - r(t)\right]^2 \right\rangle$, where $r(t)$ is the position vector of a cell at time $t$, and $\tau$ represents the time lag. The MSD provides information on the average displacement between points in the motility trajectory separated by a fixed time lag. Mean path length $\bar{l}$ was calculated as the average of the total travelling distance of each tracked cell in the field of view.

## MSHA pilus labeling, imaging, and quantification

Pilin labeling was achieved using Alexa Fluor 488 C5 Maleimide (AF488-mal; Thermo Fisher Scientific, cat. no. A10254) or Alexa Fluor 546 C5 Maleimide (AF546-mal; Thermo Fisher Scientific, cat.

no. A10258), which was dissolved in DMSO, aliquoted, and stored at −20℃ while being protected from light.

*V. cholerae* cultures were grown to mid-log phase ($OD_{600}$=0.8–1.5) before labeling. ~100 µL of culture was mixed with dye at a final concentration of 25 µg/mL (*Ellison et al., 2017*) and incubated at RT for 5 min in the dark. Labeled cultures were harvested by centrifugation (900×*g*, 5 min) and washed twice with PBS, resuspended in 200 µL PBS and imaged immediately. Images were collected using an EMCCD camera on a Leica DMi8 inverted microscope equipped with an Adaptive Focus Control system. The fluorescence of cells labeled with AF488-mal and AF546-mal was detected with FITC and Rhod filter, respectively. The cell bodies were imaged using phase-contrast microscopy.

To quantify the number of MSHA pili per cell and cell length, imaging was done under 0.2% PBS gellan gum pads. The cell lengths were measured using ImageJ.

We used AF546-mal and AF488-mal, in turn, for the two-color labeling to observe the growth of pili. We first, labeled log-phase cells with AF546-mal for the primary staining by incubating for 20 min, followed by two successive washes in PBS by centrifugation. The cells were then resuspended in LB and incubated for an additional 40 min at 30℃. For the secondary staining, we incubated the cells in AF488-mal for 5 min, washed twice with PBS, and then imaged the cells immediately using phase contrast, FITC, and RhoD channels.

## Fluorescence video acquisition of MSHA pilus-labeled cells motility in LB+MC

The labeled cells were centrifugated, resuspended in ~20 µL PBS, and then diluted in 500 µL of the viscoelastic solution of LB+MC. The solution was then immediately pipetted onto standard microscope slides. Fluorescence images were acquired at 130 ms intervals for a total of about 2–5 min. After a few minutes of fluorescence imaging, most cells in the field of view have attached to the surface, while the fluorescence was bleached due to the continuous exposure. We recorded images from different locations to capture new instances of bacterial movement and adhesion events.

## Rheological measurements

Rheological measurements of 1% MC+2% LB solution and 18.7% (w/w) mucin (Sigma, USA) in 2% LB solution were carried out on a rheometer (DHR-2, TA Instruments, Waters LLC) using a cone-plate geometry with a diameter of 40 mm and a cone angle of 5˚ at 26℃. Here, because when they were observed on a microscope, the real temperature of samples at the observation window site is lower than the set value of 30℃ due to the relatively low thermal conductivity of glass, which is estimated to be 26–28℃, the rheological measurements were performed at 26℃. The viscosity curves were determined at shear rates of 0.01–1000/s. The storage and loss moduli were measured as functions of the testing oscillation strain and oscillation frequency. For each tested solution, we left it standing for 15 min prior to each measurement to allow it to reach equilibrium, and then it was covered with a thin layer of silicone oil to prevent loss of moisture during measurement.

## Acknowledgements

The authors thank Zhanglin Hou and Thomas G Mason for their help with scientific discussions.

## Additional information

### Funding

| Funder | Grant reference number | Author |
| --- | --- | --- |
| National Key Research and Development Program of China | 2018YFA0902102 | Kun Zhao |
| National Natural Science Foundation of China | 31770132 | Zhi Liu |
| National Natural Science Foundation of China | 81572050 | Zhi Liu |
| National Natural Science | 21621004 | Kun Zhao |

| Foundation of China | | |
|---|---|---|
| University of Bristol | Vice-Chancellor's Fellowship | Rachel R Bennett |
| Japan Society for the Promotion of Science KAKENHI | 21H01720 | Andrew S Utada |

The funders had no role in study design, data collection and interpretation, or the decision to submit the work for publication.

## Author contributions

Wenchao Zhang, Data curation, Formal analysis, Investigation, Visualization, Methodology, Writing - original draft, Writing - review and editing; Mei Luo, Data curation, Formal analysis, Investigation, Methodology, Writing - original draft, Writing - review and editing; Chunying Feng, Huaqing Liu, Hong Zhang, Data curation; Rachel R Bennett, Conceptualization, Resources, Formal analysis, Supervision, Funding acquisition, Investigation, Methodology, Writing - original draft, Writing - review and editing; Andrew S Utada, Conceptualization, Resources, Supervision, Funding acquisition, Methodology, Project administration, Writing - review and editing; Zhi Liu, Conceptualization, Resources, Supervision, Funding acquisition, Investigation, Methodology, Project administration, Writing - review and editing; Kun Zhao, Conceptualization, Resources, Supervision, Funding acquisition, Investigation, Methodology, Writing - original draft, Project administration, Writing - review and editing

## Author ORCIDs

Rachel R Bennett (iD) https://orcid.org/0000-0002-6409-6967
Andrew S Utada (iD) https://orcid.org/0000-0003-4542-6315
Kun Zhao (iD) https://orcid.org/0000-0003-3928-1981

## Ethics

Animal experimentation: All mice received the humane care and the experimental protocols were carried out in accordance with the Guide for the Care and Use of Laboratory Animals, Huazhong University of Science and Technology, as approved by the Animal Care Committee of Hubei Province.

## Decision letter and Author response

Decision letter https://doi.org/10.7554/eLife.60655.sa1
Author response https://doi.org/10.7554/eLife.60655.sa2

# Additional files

## Supplementary files

• Source code 1. Source code for *Figure 5a and b* trajectory plots.

• Supplementary file 1. This supplementary information contains Table S1 which lists plasmids and primers used.

• Transparent reporting form

## Data availability

All data generated or analysed during this study are included in the manuscript and supporting files. Source data files have been provided for Figures 1–6.

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
