## [Decision Letter]

**Acceptance summary:**

The manuscript is an observational study of how MSHA pili interact with a surface to cause free swimming *V. cholerae* cells to attach and adhere, which is the initial step in biofilm formation. The present paper presents developments in labelling and characterisation of the dynamics of attachment, along with the effect of viscosity. Detailed data are presented on the location of pili, their interaction with the surface, and the associated effect on cell movement, and process is characterised through various stages from initial pili contact to final tethering.

**Decision letter after peer review:**

Thank you for sending your article entitled "Crash landing of *Vibrio cholerae* by MSHA pili-assisted braking and anchoring in a viscous environment" for peer review at *eLife*. Your article has been evaluated by 2 peer reviewers, and the evaluation has been overseen by a Reviewing Editor and Aleksandra Walczak as the Senior Editor.

As you can see from the individual reviews below, the reviewers are supportive of the experimental aspects of the work, but are concerned that there is a lack of clear message to the paper. Most significantly, there is very little detailed mechanical understanding presented of the processes at hand. The entirety of the theoretical analysis is relegated to the Methods Section and no intuitive or heuristic explanation or stringent test of the data is given. A good example of this is provided by the discussion around the persistence length of the pili. Given the emphasis on mechanical aspects of the surface interactions, we would have expected those measurements to translate into the bending stiffness of the pili and therefore some idea of the forces involved when the adhere to the surface. We are also concerned about statements such as those on lines 254+ concerning "apparent viscosities in the normal and tangential directions".

This is not a concept within standard Resistive Force Theory – there are drag coefficients in those two directions, but not differing viscosities, unless somehow one is speaking of complex fluids with internal structure.

If that is the case, then we would expect more in the way of rheological measurements than simply quoting a viscosity (i.e. one would need the shear rate dependence). Likewise, the apparent lack of consideration of near-surface hydrodynamic effects is a serious omission that requires justification before the results of RFT for these kinds of problems can be accepted.

In light of the above, we believe that the paper needs major revisions to clarify the message, to present, analyze, and discuss the data on trajectories in much greater depth, and to explain for the audience of *eLife* the theoretical analysis with much greater attention to detail.

*Reviewer #1:*

The manuscript is an observational study of how MSHA pili interact with a surface to cause free swimming *V. cholerae* cells to attach and adhere, which is the initial step in biofilm formation. The authors have previously presented data showing 'roaming' and 'orbiting' due to the synergistic effect of pili and flagellum; the present paper presents developments in labelling and characterisation of the dynamics of attachment, along with the effect of viscosity. Detailed data are presented on the location of pili, their interaction with the surface, and the associated effect on cell movement, and process is characterised through various stages from initial pili contact to final tethering. The system is examined at low and high viscosity, and it is shown that pili have a stronger effect on mean path length in high compared with low viscosity fluid. Existing mathematical fluid dynamics models are employed to interpret some of the results, for example explaining why standing cells have higher angular velocity.

1) Can any further mechanistic explanation be given for why tethering dynamics are different at high viscosity? Is it because the cells are moving more slowly?

2) A parameter estimate is given for mμ_T_* on page 15, line 260, which is presumably a fitted value. A confidence interval should be provided for this estimate. Would the 'textbook' behaviour of mμ_N_/mμ_T_ slightly less than 2 also explain the data?

3) An RFT model of motility rather simplifies the hydrodynamics of surface-swimmer interaction, in particular the reorientation of the cells produced by the hydrodynamic 'images' in the wall. In view of this, how reliable are the model trajectory predictions shown in figure 4f?

*Reviewer #2:*

The swimming trajectories of bacteria is strongly perturbed as they approach surfaces. *Vibrio cholerae* "orbit" or "roam" when swimming near a surface in a manner that depends on its MSHA pili (described in Utada et al. 2014). Here, Zhang et al. further probe the function of MSHA pili in the near surface swimming and attachment behavior of Vibrio cholerae. On the technical aspect, the authors rely on the implementation of flagellum and pili visualizations during cell landing, particularly in the context of a "viscous" environments. The paper is thus focused on improving our understanding of surface approach strategies of *V. cholerae*. The main strength of the paper is the ability to visualize MSHA pili during these events. While these visualizations are impressive and interesting, the paper lacks a clear conclusion, and in my opinion remains too descriptive. Therefore I don't think this manuscript warrant publication in *eLife*.

1. The manuscript is articulated into three parts: pili visualization, modelling using resistive theory and finally characterization of viscosity effects on trajectories. These parts are quite disjointed so that their association lacks cohesion.

2. The figures are not sufficiently meaningful to support the conclusions of the paper. Figure 1 is a quantification of pili number which feels anecdotal. Figure 2 is vague and mostly consists in showing trajectories which have been already described in previous studies. Figure 2e is interesting but barely visible, etc… To me the lack of focus of the figures is a sign that the paper lacks a clear message.

3. The RFT model fails to come up with insights in the mechanism of near surface motion.

4. As it is written, the concept viscosity is central to the paper. I assume that increased viscosity is helpful when performing pili visualizations. However, it is unclear to me whether increasing fluid viscosity truly replicates environments such as the mucus layer, which is a complex viscoelastic hydrogel and thus rheologically differs from a methylcellulose solution. Also, I want to point out that talking about changes in swimming behaviors as a function of viscosity is delicate without mention of the Reynolds number. In summary, I think the manuscript lacks a physically-oriented discussion on viscosity.

---

## [Author Response]

[…] In light of the above, we believe that the paper needs major revisions to clarify the message, to present, analyze, and discuss the data on trajectories in much greater depth, and to explain for the audience of eLife the theoretical analysis with much greater attention to detail.Reviewer #1:The manuscript is an observational study of how MSHA pili interact with a surface to cause free swimming *V. cholerae* cells to attach and adhere, which is the initial step in biofilm formation. The authors have previously presented data showing 'roaming' and 'orbiting' due to the synergistic effect of pili and flagellum; the present paper presents developments in labelling and characterisation of the dynamics of attachment, along with the effect of viscosity. Detailed data are presented on the location of pili, their interaction with the surface, and the associated effect on cell movement, and process is characterised through various stages from initial pili contact to final tethering. The system is examined at low and high viscosity, and it is shown that pili have a stronger effect on mean path length in high compared with low viscosity fluid. Existing mathematical fluid dynamics models are employed to interpret some of the results, for example explaining why standing cells have higher angular velocity.1) Can any further mechanistic explanation be given for why tethering dynamics are different at high viscosity? Is it because the cells are moving more slowly?

We had planned to get a better understanding on tethering dynamics through a model. Unfortunately, to develop a suitable hydrodynamic model that incorporates the effect of viscoelastic properties of fluid together with the mechanic properties of pili is beyond our current capabilities.

On the other hand, by testing cell motility behavior in another solution, 20% Ficoll solution which shows a high viscosity but still belongs to a Newtonian fluid (H. Winet, J. exp. Biol. 64, 283-302 (1976)), the results show that prolonged orbiting motions of △*mshA* were observed in LB+MC solutions (non-Newtonian fluid) but not in the 20% Ficoll solution (See Figure 5 in the revised manuscript). These results suggest that elastic properties of the complex fluid also play a role in affecting cell motility behavior. But to what extent that the elastic properties of the complex fluid can affect and what the role of viscosity is in the process needs further studies.

2) A parameter estimate is given for mμ_T_* on page 15, line 260, which is presumably a fitted value. A confidence interval should be provided for this estimate. Would the 'textbook' behaviour of mμ_N_/mμ_T_ slightly less than 2 also explain the data?3) An RFT model of motility rather simplifies the hydrodynamics of surface-swimmer interaction, in particular the reorientation of the cells produced by the hydrodynamic 'images' in the wall. In view of this, how reliable are the model trajectory predictions shown in figure 4f?

Since we cannot develop a suitable model for this work due to the recently measured viscoelastic properties, we have retracted all the modelling part in the revised manuscript.

Reviewer #2:The swimming trajectories of bacteria is strongly perturbed as they approach surfaces. Vibrio cholerae "orbit" or "roam" when swimming near a surface in a manner that depends on its MSHA pili (described in Utada et al. 2014). Here, Zhang et al. further probe the function of MSHA pili in the near surface swimming and attachment behavior of Vibrio cholerae. On the technical aspect, the authors rely on the implementation of flagellum and pili visualizations during cell landing, particularly in the context of a "viscous" environments. The paper is thus focused on improving our understanding of surface approach strategies of *V. cholerae*. The main strength of the paper is the ability to visualize MSHA pili during these events. While these visualizations are impressive and interesting, the paper lacks a clear conclusion, and in my opinion remains too descriptive. Therefore I don't think this manuscript warrant publication in eLife.1. The manuscript is articulated into three parts: pili visualization, modelling using resistive theory and finally characterization of viscosity effects on trajectories. These parts are quite disjointed so that their association lacks cohesion.

The focus of our work is on the pili function during cell landing. Our results show that MSHA pili act as a brake and anchor during cell landing, and this role is more apparent in viscoelastic non-Newtonian solutions than viscous Newtonian ones (Figure 5). To make this message clearer, we have added a cartoon illustration to summarize the landing process of *V. cholerae* cells and possible mechanisms associated with it ( Figure 7 in the revised manuscript), as well as modified associated discussions.

2. The figures are not sufficiently meaningful to support the conclusions of the paper. Figure 1 is a quantification of pili number which feels anecdotal. Figure 2 is vague and mostly consists in showing trajectories which have been already described in previous studies. Figure 2e is interesting but barely visible, etc… To me the lack of focus of the figures is a sign that the paper lacks a clear message.

In the revised manuscript, we have modified both Figure 1 and Figure 2. In Figure 1, we have moved one panel (panel b in the original version of Figure 1) to a supplementary figure (Figure 1—figure supplement 2). We would like to keep other panels in Figure 1 because for one thing, they are the validation of our visualization method and for another thing, the observation of evenly distributed pili along cell length with a constant length density is closely related to the later discussion on the role of pili during cell landing. In Figure 2, we have added dashed lines to indicate the envelop of cell body.

3. The RFT model fails to come up with insights in the mechanism of near surface motion.

Since we cannot develop a suitable model for this work due to viscoelastic properties, we have retracted all the modelling part in the revised manuscript.

4. As it is written, the concept viscosity is central to the paper. I assume that increased viscosity is helpful when performing pili visualizations. However, it is unclear to me whether increasing fluid viscosity truly replicates environments such as the mucus layer, which is a complex viscoelastic hydrogel and thus rheologically differs from a methylcellulose solution. Also, I want to point out that talking about changes in swimming behaviors as a function of viscosity is delicate without mention of the Reynolds number. In summary, I think the manuscript lacks a physically-oriented discussion on viscosity.

We have measured the rheological properties of MC + LB solution, and the results show that it has weak viscoelastic properties with a higher storage modulus than loss modulus, i.e., it is a non-Newtonian fluid. We have also tried to measure the rheological properties of mucin solutions with a similar concentration (18.7% w/w) which is obtained from real fresh mucus samples scraped from mouse intestine surfaces. As the Reviewer pointed, the measurement results show that these mucin solutions are also non-Newtonian fluids with a higher storage modulus than loss modulus in the same tested oscillation frequency range as for MC+LB.

To further test how the viscoelastic property of the solution affect cell behavior, we also tested 20% (w/v) Ficoll solutions, which have an increased viscosity but still behave as a Newtonian fluid (H. Winet, J. exp. Biol. 64, 283-302 (1976)). The results show that △*mshA* cells in 20% Ficoll solutions do not show the prolonged orbiting behavior as they do in 1% methylcellulose, indicating that elastic properties of the complex fluid also play a role in affecting cell motility behavior. Thus, compared with Ficoll solutions, non-Newtonian methylcellulose solutions is better to simulate the real mucus layer of intestines, as mucin solutions at a concentration comparable to the real mucus layer of intestines are also viscoelastic.

We had tried to measure the cell behavior directly either in real fresh mucus samples from mouse intestines or in mucin solutions with a similar concentration (18.7% w/w), but didn’t get meaningful cell motility data due to either poor image quality and/or sample impurities and inhomogeneity.